



# Assessing CO₂ and CH₄ fluxes from mounds of African fungus-growing termites

Matti Räsänen[1], Risto Vesala[2], Petri Rönnholm[3], Laura Arppe[2], Petra Manninen[1], Markus Jylhä[1], Jouko Rikkinen[2,4], Petri Pellikka[1], and Janne Rinne[1,5]

[1]Department of Geosciences and Geography, University of Helsinki, P.O. Box 64, 00014, Helsinki, Finland
[2]Finnish Museum of Natural History, University of Helsinki, P.O. Box 64, 00014, Helsinki, Finland
[3]Department of Built Environment, Aalto University, P.O. Box 14100, 00076, Aalto, Finland
[4]Organismal and Evolutionary Biology Research Programme, Faculty of Biological and Environmental Sciences, P.O. Box 64, 00014, Helsinki, Finland
[5]Natural Resources Institute Finland, P.O. Box 2, 00791 Helsinki, Finland

*Correspondence to*: Matti Räsänen (matti.rasanen@helsinki.fi)

**Abstract.** Termites play an essential role in decomposing dead plant material in tropical ecosystems and are thus major sources of gaseous C emissions in many environments. In African savannas, fungus-growing termites are among the ecologically most influential termite species. We studied the gas exchange from mounds of two closely related fungus-growing species (*Macrotermes subhyalinus* and *M. michaelseni*, respectively) in shaded and open habitats together with soil fluxes around the mounds. The fluxes from active termite mounds varied from 116 to 2105 mg $CO_2$-C m$^{-2}$ h$^{-1}$ for $CO_2$ and from 0.06 to 3.66 mg 20  $CH_4$-C m$^{-2}$ h$^{-1}$ for $CH_4$ fluxes. Mound $CO_2$ fluxes varied seasonally with a 64 % decrease and 35 % increase in the fluxes from dry to wet season at the open grassland and shaded bushland sites, respectively. During the wet season, the $CO_2$ fluxes were significantly correlated with termite mound volume. The diurnal measurements from two *M. michaelseni* mounds suggest that the gas fluxes peak during daytime and midnight supporting the previously presented model of thermally driven air circulation inside the mound and possibly enhanced ventilation during the night by the opening of foraging routes. The stability of the 25  mound gas fluxes over diurnal and annual scales coincides with the constant nature of the nest internal gas and thermal environment that guarantees continuously favorable conditions for the fungal symbiont. Soil fluxes of both $CO_2$ and $CH_4$ were enhanced at up to 2 m distance from the mounds compared to the local soil respiration, indicating that, in addition to mound ventilation structures, a small proportion of the produced metabolic gases leave the nest also via surrounding soils.




# 1 Introduction

Termites are major degraders of dead organic plant matter in many tropical and subtropical ecosystems (Bignell and Eggleton 2000; Jouquet et al., 2011), also making them a significant source of atmospheric carbon dioxide and methane (Zimmerman et al., 1983; Khalil et a., 1990). Especially in African dry savannas, the fungus-growing species comprise the most common and ecologically important termite group that may recycle a quarter of total dead plant matter and even 90 % of the woody litter in some arid ecosystems (Collins 1981; Buxton 1981). Fungus-growing termites cultivate symbiotic termite fungi (*Termitomyces* spp., Basidiomycota) in fungus combs, which are specialized structures within underground nest chambers. The combs are built from feces of termite workers and consist of largely undigested plant matter, initially ingested by foraging termites, that the *Termitomyces* further degrades (Rouland-Lefèvre 2000; Vesala et al., 2022a). Eventually, termite hosts get their energy and nutrition from the degraded plant matter and/or the fungal mycelium (Sieber and Leuthold, 1981; Hyodo et al., 2003; Vesala et al., 2019a).

Fungus-cultivation within termites is always associated with a complex caste system including numerous workers, soldiers, and reproductive individuals, and the total number of termites within a colony typically ranges from hundreds of thousands to up to a few million individuals (Darlington, 1986, 1990). Such large insect populations are notable sources of $CO_2$ already in themselves but, within nests of fungus-growing termites, a major proportion of the nest $CO_2$ emission is produced by the extensive fungus gardens whose total biomass in mature termite colonies may exceed 20 kilograms (Darlington, 1986; Darlington et al., 1997; Noirot and Darlington, 2000). Due to their methanogenic gut symbionts, termites are also among the major sources of atmospheric $CH_4$ although estimates about their contribution to the global $CH_4$ budged vary highly (currently considered to represent 1–4 % of the natural sources) due to uncertainties in methane-related processes and global termite biomass (Sanderson, 1996; Kirschke et al., 2013; Saunois et al., 2020). Methane can also be oxidized by soil methanotrophic microbes occurring within and around termite nests (Chiri et al., 2020). It has been evaluated that methane oxidizing bacteria can mitigate even half of the produced methane in nests some mound-building termite species (Sugimoto et al., 1998; Nauer et al., 2018).

Maintenance of large termite populations and the excessive fungal gardens necessitates effective gas-exchange and stable conditions within the nest interior. Termites of the genera *Macrotermes* and *Odontotermes* build aboveground soil mounds to stabilize temperature, maintain humidity and enhance gas exchange (Noirot and Darlington, 2000; Korb 2003, 2011). Mound structure can vary a lot depending on termite species and environmental conditions (Korb and Linsenmair 1998; Korb 2003, 2011). Even closely related species may rely on fundamentally different mechanisms of nest ventilation, as in case of Kenyan *Macrotermes michaelseni* and *Macrotermes subhyalinus* that build closed and open mounds, respectively (Darlington 1984,



1985). The open *M. subhyalinus* mounds have multiple large ventilation shafts at different elevations on the mound surface, that promote wind-induced Venturi effect, resulting in airflow through the shafts that ventilate the underground nest (Weir, 1973; Korb 2011). The closed mounds of *M. michaelseni* lack these openings and, instead, the air is circulated internally within the mound until the gas-exchange between the nest interior and ambient air takes place through the porous mound surface

(Noirot and Darlington, 2000; Ocko et al., 2017). Direct measurements of air velocities and temperatures within closed mounds have shown that air moves in convective cells following a diurnally oscillating thermal schedule (King et al., 2015; Ocko et al., 2017).

Despite the well-recognized importance of termite mounds as significant sources of greenhouse gases, field studies quantifying

gas fluxes from termite mounds are rare and often limited to species that build relatively small mounds. Previous $CO_2$ flux measurements from termite mounds have ranged from 40 to 2246 mg $CO_2$-C m$^{-2}$ h$^{-1}$ (van Asperen et al., 2021; Jamali et al., 2013; Brümmer et al., 2009). For wood-feeding termites in Australia, the seasonal variability in $CO_2$ and $CH_4$ was high with larger fluxes occurring during the wet season (Jamali et al., 2013). For soil and wood-feeding termites the relationship between mound $CO_2$ and $CH_4$ fluxes has been species-specific (Jamali et al., 2013). With the assistance of *Termitomyces* symbionts,

fungus-growing termites can utilize widely different types of plant matter including grasses, leaf litter, and dead wood (Collins 1981; Lepage 1981a) and the proportion of utilized food sources may vary both geographically and seasonally (Boutton et al., 1983; Lepage et al., 1993; Vesala et al., 2022b). Alternative food sources may also differ in their nutritional value with potential consequences to the gas fluxes. For example, a lower nitrogen content associated with grass-based compared to woody plant-based fungus combs found in Vesala et al., (2022b) could necessitate relatively higher $CO_2$ and $CH_4$ emissions, as more carbon

needs to be eliminated to gain sufficiently protein from the N deficient diet (Higashi et al., 1992; Eggleton and Tayasu, 2001). Whether such factors could affect the mound gas fluxes is currently not known.

In this study, we used static chamber method to investigate $CO_2$ and $CH_4$ fluxes originating from mounds of two common species of fungus-growing termites, *M. michaelseni* and *M. subhyalinus*, in southern Kenya. As, in the area, both species inhabit a wide range of different environments, termite mounds were measured in two contrasting habitats (open grassland and dense

bushland). In Kenyan grasslands, *M. michaelseni* mounds are surrounded by dense networks of foraging tunnels with several openings to soil surface per each m$^2$ (Lepage 1981a; Darlington 1982) that could also increase soil $CO_2$ and $CH_4$ fluxes in mound proximity. Thus, in addition to direct gas fluxes from the mounds, we also measured soil $CO_2$ and $CH_4$ fluxes around the mounds. The study objectives were: (1) to quantify the gas fluxes from the fungus-growing termite mounds with variable mound volumes and from the soil around the mounds, (2) study the interactions between the fluxes and different environmental

variables and to (3) determine the diurnal and seasonal variation in mound fluxes.



## 2 Materials and methods

### 2.1 Site description

The chamber measurements were conducted at the lowland areas surrounding Taita Hills, Kenya (Fig. 1). The mean annual

rainfall is 433 mm based on measurements at the Maktau weather station (Räsänen et al., 2020). Termite mounds were measured at bushland and grassland locations that are 15 km apart. The bushland is characterized by over 50 % of thorny shrubs and small *Acacia* spp. and *Commiphora* spp. trees (Wachiye et al., 2020). All termite mounds measured in bushland were located in shaded locations. The grassland site is located at a private game sanctuary for wildlife conservation. The vegetation is characterized by common savanna grasses such as *Chloris roxburghiana* and *Cenchrus ciliaris* and scattered

trees (e.g. *Acacia tortilis*, *Albizia anthelmintica*, *Balanites aegyptica*). Majority of bushland mounds were open mounds build by *M. subhyalinus*, whereas most of the grassland mounds were closed type build by *M. michaelseni* (Table 1, Fig. S1). The grassland ecosystem is assumed to be result from decades long wildlife conservation, which causes high population of mega-herbivore, namely elephants, which forage with leaves of the trees and bushes, thus destroying the tree cover. The tree cover and above ground biomass is much higher in the bushland areas, which are not inhabited by elephants as shown in Figure 1B

(Amara et al., 2020; Pellikka et al., 2018). In true-color Sentinel satellite image (Figure 1A), the magenta color indicate grasslands, while brownish colors indicate bushlands. Due to difference in habitat, the termites also forage differently in grasslands and bushlands, and thus may cause different fluxes.

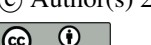



**Figure 1. Locations of the two measurement sites and weather station in Maktau in Sentinel-2A satellite image, 20 December 2020 (A) and on above ground biomass map (B). Chamber measurement at the grassland (C) and the bushland site (D). Metal collar and wireframe around an open *M. subhyalinus* mound (E). Sentinels Scientific Data HubCE4 (ESA, 2015). Photo credits: Petri Pellikka (C), and Markus Jylhä (D, E).**





**Table 1. Termite mound characteristics. R is the average mound radius, H is height and $V_m$ is calculated above ground mound volume (see section 2.2). The status (active or dead) of the colonies was determined after the second measurement campaign by**
5 **excavation.**

| Mound id | Termite species | Location | Type | $R$ (m) | $H$ (m) | $V_m$ (m³) | Status in April 2017 |
|---|---|---|---|---|---|---|---|
| MR1 | *M. michaelseni* | Bushland | closed | 0.75 | 0.50 | 0.256 | active |
| MR2 | *M. subhyalinus* | Bushland | open | 0.65 | 0.15 | 0.058 | active |
| MR3 | *M. subhyalinus* | Bushland | open | 0.28 | 0.33 | 0.023 | dead |
| MR4 | *M. subhyalinus* | Bushland | open | 0.75 | 0.45 | 0.231 | active |
| MR5 | *M. subhyalinus* | Bushland | open | 0.30 | 0.25 | 0.020 | active |
| MR6 | *M. subhyalinus* | Bushland | open | 0.40 | 0.30 | 0.044 | active |
| S1 | *M. michaelseni* | Grassland | closed | 0.75 | 0.70 | 0.259[a] | active |
| S2 | *M. michaelseni* | Grassland | closed | 0.65 | 0.35 | 0.106[a] | active |
| S3 | *M. michaelseni* | Grassland | closed | 0.55 | 0.30 | 0.217[a] | dead |
| S4 | *M. michaelseni* | Grassland | closed | 0.35 | 0.40 | 0.074[a] | dead |
| S5[b] | *M. subhyalinus* | Grassland | open | 0.75 | 0.40 | 0.193[a] | active |
| S6 | *M. subhyalinus* | Grassland | open | 0.72 | 0.15 | 0.072 | dead |
| | [a] These values were estimated using photogrammetric technique.<br>[b] The gas fluxes of this mound were not estimated due to nonlinear concentration time series (Fig. S2) | | | | | | |

## 2.2   Estimation of mound volume

Volumes of the grassland mounds S1–S5 (Table 1) were measured photogrammetrically by taking a large number of digital images from different positions around the mound (method described in Vesala et al., 2019b). To increase the number of observations, this data was supplemented with five additional comparable-sized mounds measured earlier in a close by area
(Vesala et al., 2019b). Linear regression with zero intercept was fitted between equivalent cone volume and photogrammetric mound volume (Fig. 2). This relationship was used to estimate the volume at the bushland site where the photogrammetric method was not feasible due to the surrounding dense vegetation. The equivalent cone volume was calculated as

$$V_{cone} = \frac{1}{3}\pi R^2 \times H, \tag{1}$$

where $R$ is the average of North-South and West-East measured radii and $H$ is height. The above ground termite mound volume ($V_m$) was $0.87 \cdot V_{cone}$ for mounds that were not sampled using the photogrammetric method. The above ground termite mound volumes ranged from 0.3 to 10 % of the total chamber volume.

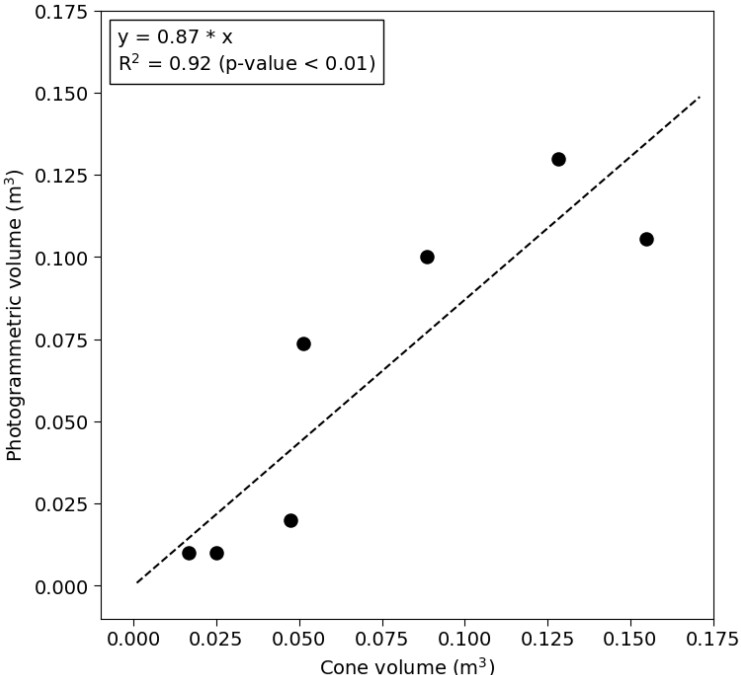

**Figure 2. The relationship between the photogrammetrically measured mound volume and calculated cone volume according to equation 1. The data points include grassland mounds (S2 and S4) of this study and five additional mounds measured in Vesala et al., (2019b) (Table S1).**

### 2.3    Gas flux measurements from termite mounds and soil

$CO_2$ and $CH_4$ fluxes were measured using a static chamber method in November 2016 (end of long dry season) and April 2017 (wet season). The chamber collar was installed before each measurement and surrounding sand was applied to insulate collar. The chamber closure time was 15 minutes, and each mound was measured three times with five-minute breaks in between the

10    repetitions. Daytime flux measurements were made between 10 am and 4 pm. To study diurnal variation in gas fluxes, additional nighttime measurements were made at two closed mounds (S1 and S2).

The size of the measurement chamber was a compromise between practicality and typical size of termite mounds in the area. The radius of the chamber was 0.75 m and height was 1.5 m, still excluding the vast majority of *Macrotermes* mounds occurring

15    in the area, which are larger on average. The metal collar height was 40 cm. Four upright positioned steel rods and an attaching circular steel frame were used to support the cylindrical shape of the custom made polyvinylchloride sheet. A rubber band was used to seal the polyvinylchloride sheet against the collar. Three people were needed to open and close the chamber. The gas analyzer (LGR Ultra-Portable Greenhouse Gas Analyzer Model 915-0011) was connected to the collar using Swagelok fittings. Chamber air was sampled continuously into the gas analyzer and circulated back to the chamber. The air was sampled inside





the chamber using a 2.5 m long teflon tube which had small holes along its length. The sampling line was attach in a spiral shape by supporting it from the middle points of the supporting steel rods and it was connected to the inlet of the gas analyzer at the inside wall of the collar. The air inside the chamber was mixed using three 12 cm fans that were hanging in the middle point of the chamber. The gas analyzer recorded concentration every second. Temperature and relative humidity inside the

chamber was measured using a Campbell CS215 T/RH sensor attached to the collar. A cellphone timing application (Time meter) was used to record the precise start and end times of each measurement because the gas analyzer produces one continuous measurement file per day. Also, the manual soil moisture and temperature measurements were noted in the timing application. The notes were exported to CSV file which was parsed into numerical form. This procedure enabled to automate flux calculation from raw data and minimize note taking errors.

After termite mound measurement, the inlet lines were connected to a small opaque soil chamber (radius 7 cm, height 22 cm) to measure soil $CO_2$ and $CH_4$ fluxes around the termite mound. The soil gas flux measurements were made once for two opposing directions from 2, 4 and 6 m distance from the perimeter of the mound. The soil chamber was placed on patches of bare soil to minimize the effect of vegetation. The sampling time was two minutes. One $CH_4$ flux measurement around S1

mound at 6 m distance during wet season was discarded because the flux was over thousand times higher than the mean of other measurements around the mound (most likely due to an underground termite nest).

Methane and carbon dioxide fluxes were calculated as

$$F = \frac{PV}{RTA}\frac{dC}{dt},\qquad\qquad(2)$$

where $V = V_c - V_m$ for termite mound measurements and $V = 0.0034$ m$^3$ for the soil measurements, $V_c = 2.65$ m$^3$ is the termite chamber volume, $P$ is pressure, $A$ is the area of the chamber, $R$ is gas constant ($8.314\ Jmol^{-1}K^{-1}$) and $T$ is air temperature in Kelvin. The slope of concentration increase $\frac{dC}{dt}$ was determined from a linear regression of the 1 Hz concentration measurements.

### 2.4   Environmental measurements

The soil moisture and temperature were measured manually nearby the mound during each chamber measurement. A handheld soil moisture probe Campbell HydroSense II was used to measure surface soil moisture. Soil temperature was measured using a stainless-steel temperature probe. Termite nest temperature measurements were used from an earlier study (Vesala et al., 2019b) where small temperature sensors (iButton Thermochron DS1922L) were placed next to the first fungus combs that were encountered by digging small hole to the mound. The nest temperature measurements made at the grassland location in

2015 from large open and closed mounds (IDs TS56 and TS19 in Vesala et al., 2019b) were used. These nest temperatures were compared to continuous soil temperature measurements at the Maktau weather station. The weather station soil temperature (Campbell CS650) probe was measuring at 30 cm depth inside a fenced area.





The $CO_2$ and $CH_4$ fluxes measured for each mound were correlated with above ground biomass (AGB) assessed using airborne laser scanner data and field measurements (Amara et al., 2020; Pellikka et al., 2018) applying the logic that fluxes may vary with the amount woody vegetation in the environment (Vesala et al., 2022b).

**2.5    Statistical analysis**

Simple linear regression was used to test the statistical relationship between the measured fluxes and explanatory variables. The relationship between fluxes and mound volume was tested separately during the dry season (n=11) and wet season (n=7). For the relationship between environmental variables and fluxes and the ratio of $CH_4$ to $CO_2$ flux was tested using all the measurements (n=18). The mound fluxes and AGB relationship was tested with simple linear regression during the dry season (n=11). Statistical analyses were done using the Scipy Python package (Virtanen et al., 2020).

**3    Results**

**3.1    Environmental conditions**

Total precipitation before the first measurement campaign was 10 mm from September to early November (Fig. 3), meaning that the short rainy season had not yet started. The precipitation between the campaigns was 153 mm corresponding to a typical dry year amount. The annual mean precipitation during this period between the campaigns was 287 mm with 154 mm standard deviation calculated from six years of precipitation measurements at the weather station. The wind speed had a consistent trend during the measurement campaign days showing a minimum around 6 am and a maximum during the early afternoon. Revisiting the data collected in Vesala et al. (2019b) shows that termite mounds in the area have a consistent diurnal temperature rhythm with the maximum nest temperature being always around 8 pm, whereas the soil temperature at 30 cm depth has a maximum around midnight.





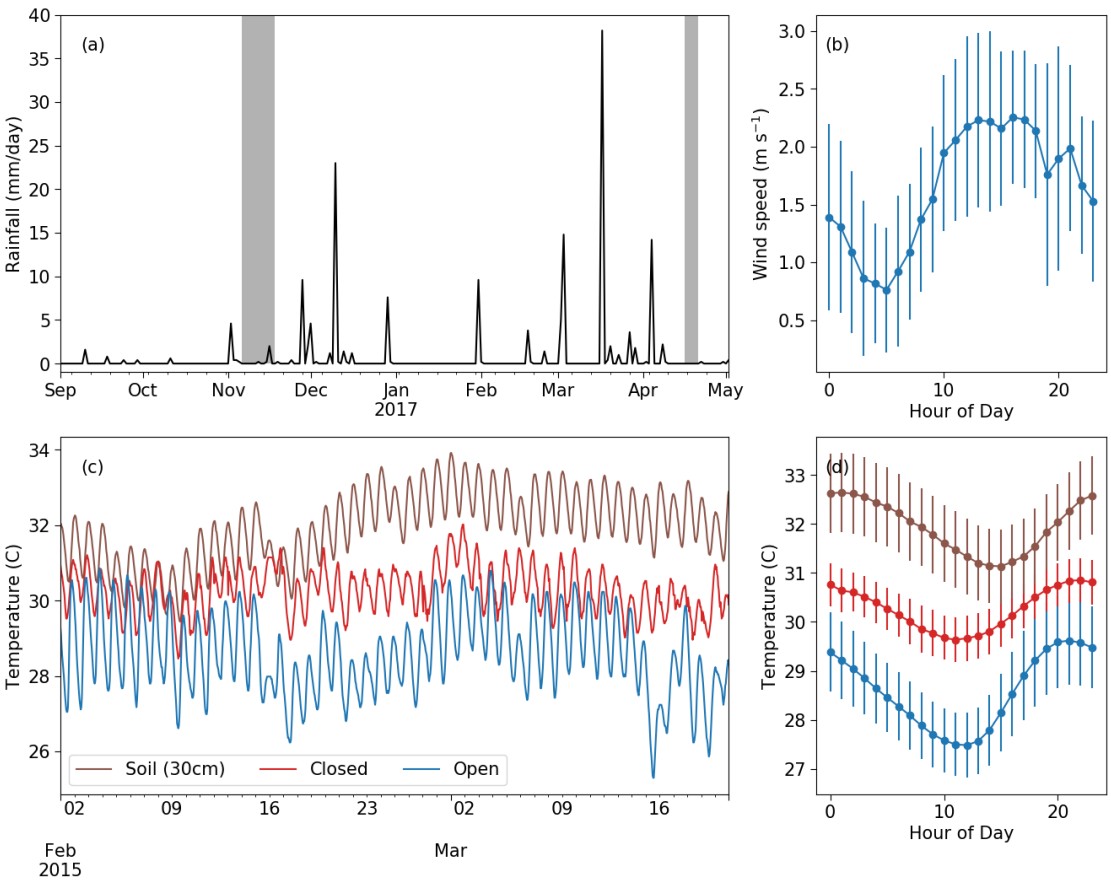

**Figure 3. (a) Time series of daily rainfall. The gray areas indicate the days of the measurement campaigns. (b) Diurnal mean of the wind speed during the first measurement campaign. (c) Soil temperature at 30 cm depth and termite nest temperatures for open (TS56) and closed (TS19) mounds at the grassland site in 2015 during period of no rainfall (Vesala et al., 2019b). (d) Diurnal mean of the soil and nest temperatures during the same period. The error bars indicate one standard deviation.**

## 3.2    Mound CO₂ and CH₄ fluxes

The mean $CO_2$ fluxes ranged from 35 to 2105 mg $CO_2$-C m$^{-2}$ h$^{-1}$ for closed *M. michaelseni* mounds and from 116 to 809 mg $CO_2$-C m$^{-2}$ h$^{-1}$ for open *M. subhyalinus* mounds (Fig. 4). The mean $CO_2$ flux was 1047 mg $CO_2$-C m$^{-2}$ h$^{-1}$ for the grassland site and 380 mg $CO_2$-C m$^{-2}$ h$^{-1}$ for the bushland site from the active mounds listed in Table 1. The largest $CO_2$ fluxes were measured during the dry season at the grassland site. The site level mean $CO_2$ flux decreased 64 % at the grassland and increased 41 % at the bushland from dry to wet season. A comparison of the closed mound S1 and open mound MR4, which have similar volumes (Table 1), shows that the closed S1 mound had higher $CO_2$ and $CH_4$ flux than the open MR4 mound during the dry season (Fig. 4). The lowest $CO_2$ fluxes were registered from the grassland mounds S3 and S4, which turned out to be dead when opened after the second measurement campaign (Table 1).





The mean $CH_4$ fluxes ranged from 0.02 to 3.66 mg $CH_4$-C $m^{-2}$ $h^{-1}$ for closed *M. michaelseni* mounds and 0.06 to 1.15 mg $CH_4$-C $m^{-2}$ $h^{-1}$ for open *M. subhyalinus* mounds. The ratio of mound $CH_4$ flux to mound $CO_2$ flux ranged from $0.4 \cdot 10^{-3}$ to $2.8 \cdot 10^{-3}$ with no clear difference between termite species or mound volume. The mean ratio was $1.6 \cdot 10^{-3}$ for grassland and $1.2 \cdot 10^{-3}$ for bushland sites. The ratio of $CH_4$ flux to $CO_2$ flux reduced 36 % at the grassland and 49 % at the bushland site

5     from dry to wet season. The dry season mound $CO_2$ and $CH_4$ fluxes were not significantly correlated with the AGB (Fig. S3).





Figure 4. Mean termite mound $CO_2$ and $CH_4$ fluxes from grassland and bushland sites during dry and wet seasons. Each mound was sampled once in each season during daytime (10 am to 4 pm) by three repeated measurements. Error bars indicate standard error. Hollow bars indicate open and solid bars closed mounds. The mounds that were dead in April 2017 were only measured during the dry season. The mounds are presented in order of increasing volume for each site.

Based on diurnal measurements in two closed mounds, the largest $CO_2$ flux was measured around noon at the mound S1 (Fig. 5) coinciding with the maximum diurnal wind speed and the minimum nest temperature (Fig. 3). However, the $CO_2$ flux increased also during the midnight for both S1 and S2 mounds. For both mounds, the difference between maximum nighttime $CO_2$ flux compared with daytime maximum was less than 2 standard errors. For S1 mound, the maximum flux value was 1.4 times the daily average for $CO_2$ and 1.5 times for $CH_4$. The nighttime peak in the $CO_2$ flux coincides with the maximum soil temperature at 30 cm depth at midnight (Fig. 3d). The diurnal cycle of the ratio of $CH_4$ to $CO_2$ flux showed one peak around the noon for S1 mound and at 6 pm for S2 which was not sampled during the noon. These peaks are due to the increase in $CH_4$ flux in proportion to $CO_2$ flux during the daytime.

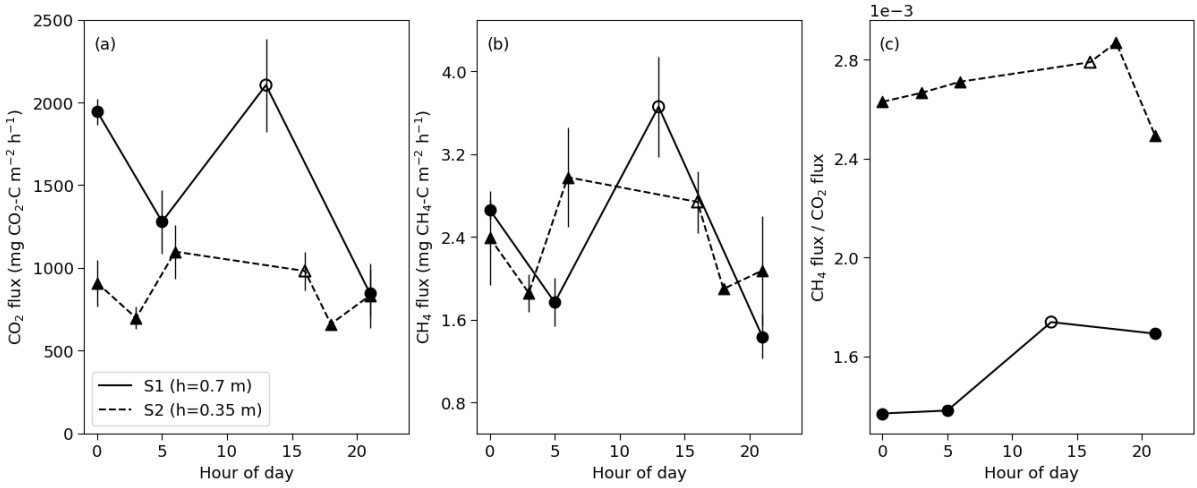

**Figure 5. Diurnal variation of $CO_2$ and $CH_4$ fluxes during 2016 dry season for closed mounds S1 and S2. Error bars indicate standard error. The hollow symbols indicate daytime measurements that were done nine days before nighttime measurements for S1 and one day before for S2.**

### 3.3 Drivers of mound $CO_2$ and $CH_4$ fluxes

The mean mound $CO_2$ flux was significantly correlated with the termite mound volume during the wet season ($R^2 = 0.96$, p-value < 0.01; Fig. 6b), while the relation was scattered during the dry season (Fig. 6a). The $CO_2$ flux and $CH_4$ flux were not explained by surface soil moisture or soil temperature during the dry or wet season (Fig. S4). The soil moisture values were low for most mounds due to low precipitation amount during the wet season. The mean $CO_2$ and $CH_4$ flux were linearly related (Fig. 6c). The ratio of $CH_4$ and $CO_2$ fluxes had no significant correlation with environmental variables and only a weak correlation with the magnitude of $CH_4$ flux ($R^2 = 0.34$, p-value = 0.01). The $CH_4$ flux was related to the mound volume during the wet season with slightly more scatter than the equivalent $CO_2$ relation (Fig. 6e). The standard deviation of $CO_2$ flux was increasing linearly with the mean flux magnitude (Fig. 6f).

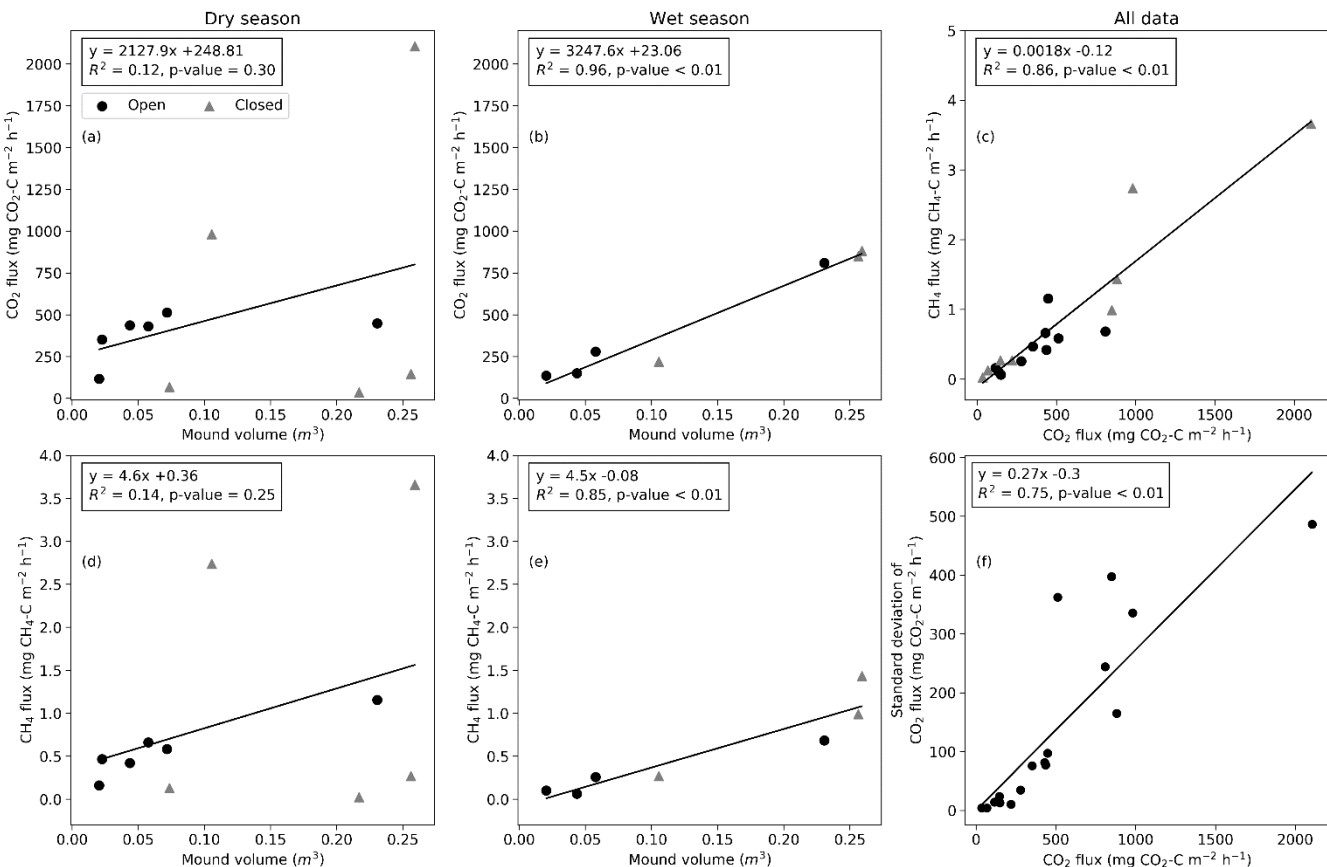

**Figure 6.** The relationship between mound volume and mean $CO_2$ and $CH_4$ fluxes during dry and wet seasons (a-b and d-e). (c) The relationship between $CH_4$ and $CO_2$ fluxes. (f) The relationship between standard deviation and mean of the $CO_2$ flux.

### 3.4    Soil $CO_2$ and $CH_4$ fluxes around termite mounds

The soil $CO_2$ and $CH_4$ fluxes around the mounds were sampled to determine the effect of termites on soil fluxes around the mound. The site mean soil $CO_2$ and $CH_4$ were always higher at a 2 m distance compared to a 4 and 6 m distance from the perimeter of the mound (Fig. 7 and Table 2). The mean soil $CO_2$ flux at the bushland was only 29 % of the mean soil flux at the grassland. The dry season means of all soil $CO_2$ flux measurements were 62 mg $CO_2$-C m$^{-2}$ h$^{-1}$ and 27 mg $CO_2$-C m$^{-2}$ h$^{-1}$ at 2 m and 4 to 6 m distance (Table 2). The difference between soil $CO_2$ flux at 2 m and 4 to 6 m distance from the mound was less pronounced during the wet season. The mean mound $CO_2$ fluxes were a factor of 25 and 6 higher at the dry and wet seasons in comparison to the 4 to 6 m distance soil $CO_2$ fluxes (Table 2). The dry season means of all soil $CH_4$ flux measurements were 0.065 mg $CH_4$-C m$^{-2}$ h$^{-1}$ and 0.020 mg $CH_4$-C m$^{-2}$ h$^{-1}$ at 2 m and 4 to 6 m distance (Table 2). The dry season mean soil $CH_4$ flux was positive at the grassland and bushland site, whereas the mean wet season flux was nearly zero




with most fluxes being negative (Fig. 7, S5 and S6). The soil $CH_4$ fluxes at a 2 m distance from the mound are much higher in proportion to fluxes at a 4 m distance at the grassland compared to the bushland (Fig. 7).

**Table 2. Mean $CO_2$ and $CH_4$ fluxes of termite mounds and surrounding soils. Data from all active mounds measured at grassland and bushland. The range of measured values for each group is given in the parenthesis.**

|  | $CO_2$ flux (mg $CO_2$-C m$^{-2}$ h$^{-1}$) | $CH_4$ flux (mg $CH_4$-C m$^{-2}$ h$^{-1}$) |
|---|---|---|
|  | Dry season | |
| Termite mound | 666 (116–2105) | 1.294 (0.159–3.659) |
| Soil at 2 m distance | 62 (10–122) | 0.065 (0.001–0.186) |
| Soil at 4–6 m distance | 27 (6–123) | 0.020 (–0.003–0.108) |
|  | Wet season | |
| Termite mound | 475 (136–881) | 0.542 (0.062–1.433) |
| Soil at 2 m | 90 (29–243) | 0.006 (–0.034–0.104) |
| Soil at 4–6 m | 83 (15–251) | –0.004 (–0.032–0.065) |

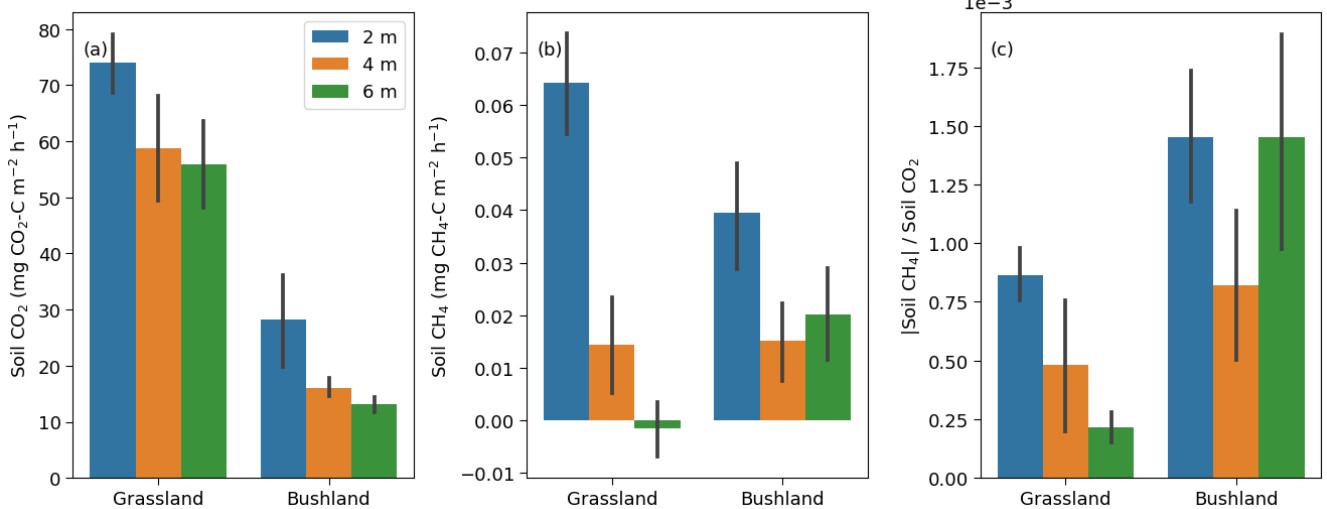

**Figure 7. (a-b) Mean soil $CO_2$ and $CH_4$ fluxes around the termite mounds for 2, 4, and 6 m distances from the perimeter of the mound during the dry season. The mean is calculated from all the mounds at the site. Soil around each mound was measured at three different distances and from two different directions. Error bars indicate standard error. (c) The ratio of absolute value of soil $CH_4$ and $CO_2$ flux.**



## 4    Discussion

### 4.1    Mound CO$_2$ and CH$_4$ fluxes

The measured CO$_2$ fluxes (from 35 to 2105 mg CO$_2$-C m$^{-2}$ h$^{-1}$) were in similar range as reported in two earlier comparable studies where chamber measurements have been used to study greenhouse gas emissions from termite mounds in Australia

(up to 1550 mg CO$_2$-C m$^{-2}$ h$^{-1}$, Jamali et al., 2013) and Amazonia (192–2246 mg CO$_2$-C m$^{-2}$ h$^{-1}$, van Asperen et al., 2021). Methane fluxes ranged from 0.02 to 3.66 mg CH$_4$-C m$^{-2}$ h$^{-1}$ and were 2–20 times lower than those reported in earlier studies (0.4–6 mg CH$_4$-C m$^{-2}$ h$^{-1}$, van Asperen et al., 2021; Jamali et al., 2013). As in earlier chamber measurement studies termites have been other than fungus-growing species, this result was expected: in fungus-growing termites a large proportion of the mound total CO$_2$ fluxes originate from aerobic fungus gardens which seem not to emit measurable amounts of CH$_4$ (Darlington

et al., 1997).

In both *Macrotermes* species studied here, mound outer dimensions correlate positively with the size of the termite population and the total biomass of fungus combs (Darlington and Dransfield 1987; Darlington 1990). Thus, the fluxes of metabolic gases should logically also show correlation with the mound volume.  However, considering the high variation associated with the

fluxes, it seems obvious that if the sizes of the studied mounds are too close to each other, the relationships between mound volume and gas fluxes are difficult to detect. In this study, the mound volumes ranged from 0.023 to 0.259 m$^3$ (Table 1) including notably more size variance than the earlier comparable studies (0.02 to 0.077 m$^3$) (Jamali et al., 2013; van Asperen et al., 2021). We found that the measured CO$_2$ fluxes were linearly correlated with the termite mound volume during the wet season (Fig. 6b) but the relationship was more indistinct during the dry season (Fig 6a). Determining the relationship between

mound volume and gas fluxes more reliably would require measurements from still much larger mounds, which however is problematic due to technical limitations. In East Africa, average volumes of *Macrotermes* mounds range from 2 to 3 m$^3$ depending on the geographical area, and the maximum volumes can exceed 12 m$^3$ (Pomeroy, 1977). Also in our study area, *M. subhyalinus* mounds with volumes of up to 6 m$^3$ have been recorded (Vesala et al., 2019b), which is more than 20 times the volume of the largest mound measured here. Thus, the reported average flux values in this study are representative for the

smallest *Macrotermes* mounds but when scaling the results to the level of larger *Macrotermes* mounds (or calculating area-based estimates), the effect of mound volumes should be considered to avoid serious underestimations.

The mounds at the grassland and bushland sites had contrasting seasonality of CO$_2$ fluxes with 64 % decrease at the grassland and 35 % increase at the bushland site from dry to wet season (Fig. 4). This seasonal change is smaller than what has been

measured for grass and wood-feeding termites in Australia, which had a 90 % decrease in CO$_2$ flux from wet to dry season (Jamali et al., 2013). The relatively small intra-annual changes found between our two measurement campaigns reflects the fact that sterile populations within *Macrotermes* nests remain relatively constant throughout the year (Darlington 1986, 1991). However, some intra-annual variation can be caused by biomass changes in other nest components. For example, within



Kenyan *Macrotermes michaelseni* nest living biomass can be briefly elevated by the annual brood of alates reaching maximum just before swarming in October–November (Darlington 1986). Correspondingly, storages of fungus combs are typically at their largest soon after the long rains in May–June after which they start to decline as food availability decreases and, concurrently, the nutritional resources are increasingly allocated to fat accumulated in developing alates (Lepage 1981b).

Additionally, differences in food quality could potentially affect nest CO2 emissions. Within our study area, fungus combs of grassland mounds had significantly lower nitrogen contents than those situated in more wooded habitats suggesting that, on average, grass-based diets have lower nutritional value than wood-based diets (Vesala et al., 2021b). Such differences in food nutritional quality may reflect to nest CO2 fluxes as more carbon needs to be eliminated to accumulate enough nitrogen from low-nitrogen food sources compared to those with a higher nitrogen content (Higashi et al, 1992). Most likely, food nutritional

quality also changes during the year, with the changes possibly being more dramatic in grasslands having fewer and more competed food sources available for termites than the more diverse bushlands. Such local differences in food availability and quality could potentially lead to different seasonality in nest living biomass which might explain the opposite seasonal changes in $CO_2$ fluxes between grassland and bushland mounds.

There was a clear decrease in $CH_4$ to $CO_2$ flux ratios from dry to wet season measurements. This difference could be most logically linked to the activity of methanotrophic bacteria within mound structures and/or surrounding soils that have been reported to consume a notable proportion of $CH_4$ produced in nests of several mound-building termite species (Sugimoto et al., 1998; Nauer et al., 2018). During wet seasons, soils and mound structures have higher water content than during dry periods, which most likely enhances the activity of the soil microbiome including the methane-oxidizing bacteria. This result

is also in congruence with our measurements of nest internal gas concentrations within *M. michaelseni* mounds, which tended to have higher $CH_4$ to $CO_2$ concentration ratios during the driest period compared to seasons with higher soil moisture (Vesala et al., in preparation).

Four mounds that were measured during the first campaign were determined to be dead at the end of the second campaign

(Table 1). The exceptionally low $CO_2$ fluxes measured from the mounds S3 and S4, however, strongly suggest that these two mounds were dead, or at least seriously weakened, already during the first measurement campaign. An average lifespan of a *Macrotermes bellicosus* colony is thought to be around four years, whereas the mounds that they build persist typically much longer (Pomeroy, 1976). However, a recent long-term monitoring study in Namibia found that 57% of initially active *M. michaelseni* mounds were still active after 12 years (Wildermuth et al., 2022), demonstrating that some colonies can live much

longer. These authors also found that small mounds were more likely to be dead than large mounds after 12 years, showing that large well-established colonies are less vulnerable to disruptions. Hence, dead *Macrotermes* mounds are a common feature in many landscapes and for example, in many places within our study area, their number exceeds that of the active mounds (Vesala et al., 2017). Thus, determination of the proportion of active mounds in a landscape is a necessary precondition for ecosystem-level termite flux estimations for all such termite species that build persistent soil mounds.



## 4.2 Diurnal variation

Diurnal variation of $CO_2$ and $CH_4$ fluxes were studied in two closed *M. michaelseni* mounds. The afternoon peaks were on average 46 % and 57 % higher compared to the daily averages for $CO_2$ and $CH_4$ fluxes, respectively (Fig. 5). Our observations are in congruence with the nest internal $CO_2$ concentration measurements by Ocko et al. (2017) who found that $CO_2$ levels within *M. michaelseni* nests in Namibia peaked in early afternoon, whereas concentrations in night-time and especially soon after sunrise were somewhat lower. A corresponding trend has been documented also from outflow chimneys of Kenyan *Macrotermes jeanneli* that had the highest $CO_2$ concentrations at 3 pm and the lowest around 3–6 am (Darlington et al., 1997). The daily maximum $CO_2$ flux in early afternoon found in this study (Fig. 5) agrees well with these earlier data suggesting that the internal $CO_2$ concentrations within *M. michaelseni* mounds show similar diurnal patterns to the mound $CO_2$ fluxes. A positive relationship between the nest internal gases and mound fluxes has been demonstrated for a few of mound-building termites (Khalil et al., 1990; Jamali et al., 2013). In agreement with our $CH_4$ measurements, a similar peak in $CH_4$ fluxes soon after midday was documented by Jamali et al. (2011) from mounds of three Australian termite species.

The lower $CO_2$ and $CH_4$ fluxes during nighttime compared to daytime fluxes are probably mostly associated with the diurnal movements of termites: after sunset a proportion of termite workers and soldiers leave the nest due to foraging activities, thus decreasing the $CO_2$ and $CH_4$ production within the nest. During night, termites also open the numerous foraging passages that directly connects the nest interior space to the soil surface near the mound (Lepage 1981a; Darlington 1982). This might lead in enhanced nest ventilation and could additionally affect gas fluxes originating from the mound. In addition to the afternoon flux peaks, we registered another peak in both $CO_2$ and $CH_4$ fluxes at midnight, which has not been documented in earlier studies (Fig. 5). Reason for this midnight peak in gas fluxes is not currently understood but might be related to temperature gradient driven diurnal changes in mound internal air currents that slow down and may even shift their direction as the mound surface cools down after sunset (Korb and Linsenmair 2000; Ocko et al., 2017). Temporary decreases in nest internal gas flows in the evening could lead in accumulation of nest internal gas concentrations (Korb, 2011) which, in turn, might trigger slightly enhanced $CO_2$ and $CH_4$ fluxes soon after the internal flows are reverted again.

In addition to the $CO_2$ and $CH_4$ fluxes, also their mutual relationship showed a clear diurnal pattern. The mound $CH_4$ to $CO_2$ flux ratio was higher during the daytime than during the nighttime measurements (Fig 5c). This result also supports the interpretation that termite population within the mound is smaller during nighttime than during daytime due to foraging activity. Termite metabolism produces both $CH_4$ and $CO_2$, but a major proportion of $CO_2$ originates from the fungal gardens, which are sessile and do not produce $CH_4$ (Darlington et al., 1997). Thus any changes in termite biomass within the nest should inevitably reflect to the mound $CH_4$ to $CO_2$ flux ratios.



After all, the diurnal changes in gas fluxes measured in this study were relatively small. This underlines the fact that plant biomass degradation within mounds of fungus-growing termites is a continuously ongoing process and show only minor diurnal changes. Therefore, also a large proportion of termites stay constantly in the nest taking care of the fungus gardens and thus the diurnal biomass changes within the nest are relatively small. The stable diurnal flux variability is in line with the

stable humidity and temperature conditions within mounds of fungus-growing termites. Constant temperature and high humidity are of key importance for the fungal *Termitomyces* symbionts which have relatively narrow range of optimal microclimatic conditions (Lüscher, 1961; Thomas 1981; Wood and Thomas, 1982).

### 4.3 Soil $CO_2$ and $CH_4$ fluxes around the mounds

The soil $CO_2$ and $CH_4$ fluxes were always higher close to the mound (at 2 m distance from the mound perimeter) compared to

further distances (4 to 6 m from the mound perimeter) (Fig. 7). The mean soil $CO_2$ flux from 4 to 6 m distance was 27 mg $CO_2$-C m$^{-2}$ h$^{-1}$ during the dry season, which is comparable to the measured soil $CO_2$ flux seasonal minimum of 20 mg $CO_2$-C m$^{-2}$ h$^{-1}$ at the grassland site (Wachiye et al., 2020). Correspondingly, the mean soil $CH_4$ fluxes at 4 to 6 m distances from the mounds were 0.020 and -0.004 mg $CH_4$-C m$^{-2}$ h$^{-1}$ at dry and wet seasons, respectively (Table 2), which is similar to the earlier field study where the soil $CH_4$ fluxes were slightly negative with no seasonality at the grassland site (Wachiye et al., 2020).

Our results thus demonstrate that, although the majority of the gas-exchange between termite nests and ambient air takes place through the mound structures, a small proportion of gases leave the nest also via adjacent soils. These results differ from the earlier measurements in soil feeding termites in Amazon which had enhanced soil fluxes around the mound only up to 0.5 m from the perimeter of the mound (van Asperen et al., 2021). The difference can be largely explained by the much larger size of the studied mounds in our study but it most likely reflects also some fundamental differences between the mounds of soil-

feeding and fungus-growing termites. While within soil-feeding species the mound itself acts as a nest for the colony, mounds built by many fungus-growing species are more or less empty from the insects and the actual nests are largely located underground below the mounds (Korb, 2011). Thus, the actual nests of fungus-growing termites may be much larger and extend to a somewhat wider area than their aboveground mounds. *Macrotermes* termites also have an extensive network of foraging tunnels surrounding their nest with the largest radial tunnels having width of 4–7 centimeters (Darlington, 1982).

Such structures increase soil porosity around the mounds enhancing water infiltration (Jouquet et al., 2011) and, obviously, also enable some gas-exchange to occur through the soil.

We found that $CH_4$ fluxes at 2 m distance from the mounds were more enhanced (compared to further distances) at the grassland than in the bushland (Fig. 7). This might simply reflect the size of the underground nest, which may have extended

to a larger area in grassland than in bushland, but might also be affected by the fact that in grasslands termites have more homogenously distributed food sources (i.e., uniform grass layer) compared to the randomly distributed food resources available at the bushland site (e.g., fallen branches and logs of dead wood). Optimal exploitation of such differently distributed food sources might lead termites to adjust their foraging networks in different habitats, with grasslands potentially having more

densely branched tunnel network with more densely distributed foraging passages in mound surroundings than in bushlands. This, in turn, might lead to a more porous soil structure in the vicinity of grassland mounds. Such interactions are, however, currently poorly known.

## 5    Conclusions

The $CO_2$ and $CH_4$ fluxes from fungus-growing termite mounds show diurnal and seasonal variation, albeit lower in magnitude than their wood, grass, and soil-feeding counterparts. The likely reason for this stability is the needed temperature and humidity control for the fungal symbiont. The $CO_2$ fluxes scale with mound volume over a large volume range. The soil flux measurements around the mound also show that the soil $CO_2$ and $CH_4$ fluxes are significantly affected by the termites at a 2 m distance from the perimeter of the mound. The fungus-growing termite mounds have lower $CH_4$ fluxes per unit $CO_2$ than soil

and grass-feeding termites, which is due to the plant matter decomposition by *Termitomyces* fungi that do not emit $CH_4$. For upscaling the greenhouse gas fluxes originating from mounds of fungus-growing termites, the work highlights the need to determine the proportion of active mounds in the landscape.

*Data availability.* The data used in this study are available online: https://doi.org/10.6084/m9.figshare.21739484.v1 (Räsänen et al., 2022).

*Author contributions.* MR, PM, MJ, PP, JR conducted the measurements; MR, RV, PR, LA, PP, JR designed the analysis; All authors contributed to the final version of the manuscript.

*Competing interests.* The authors declare that they have no conflict of interest.

*Acknowledgments.* This work was supported by the Academy of Finland through the projects SMARTLAND (grant number 318645)— Environmental sensing of ecosystem services for developing climate smart landscape to improve food security in

East Africa (PI P. Pellikka) and IsoTermes (grant number 333868). We thank Venla Pellikka, Mwadime Mjomba, Muhia Gicheru and Darius Kimuzi from Taita Research Station of the University of Helsinki and Remig Righa Mjomba, Wilson Mkala and Tobius Mwamburi from Taita Hills Wildlife Sanctuary for the support during the field work campaign. Research permit from the National Council for Science and Technology, Kenya, permit N° NCST/RCD/17/012/33 is appreciated.




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
