# Peer review of "Carbon dioxide and methane fluxes from mounds of African fungusgrowing termites"

_Biogeosciences, 2023_

## Author Comment (AC2)

page 3, line 17: for information on feeding guild and termite methane see Zhou, Yong ,A. Carla Staver, and Andrew B. Davies. 2023. Species-Level Termite Methane Production Rates. Ecology 104(2): e3905. https://doi.org/10.1002/ecy.3905

Thank you. We have added this reference to the sentence.

page 4, line 16: how are foraging strategies different here? Is it about what the termites are consuming, and why might different fluxes result?

We have removed this sentence as it's not appropriate point for site description.

page 15: for mean $CO_2$ and $CH_4$ fluxes of mound and soils, are soils significant? It looks like methane in soils is negligible.

The dry season soil $CH_4$ fluxes are positive at 2 m and 4-6 m distance. In contrast, the soil $CH_4$ fluxes far from termite mounds are negative throughout the year in this area (Wachiye et al., 2020). We were directly sampling the $CH_4$ concentrations every second and the soil $CH_4$ concentrations showed a clear linear trend that was higher than the gas analyzer noise.

page 15 figure 7: might be clearer to visualise this with mound and soil fluxes compared instead of just presenting soil fluxes, as this contextualizes both measurements.

Thank you for the suggestion. We have now added the mean mound fluxes to the figure.

page 16, line 8: Fungus-farming termites are thought to have higher rates of methane production: see Rouland et al. 1993, Gomathi et al 2009, Zhou et al 2023.

We edited the sentence and included the suggested information that most studied fungus-growing species have relatively high $CH_4$ production rates.

page 18, line 26: Sentence doesn't quite make sense - "In addition to the daily cycles of $CO_2$ and $CH_4$ fluxes, the relationship between $CO_2$ and $CH_4$ showed a clear diurnal pattern"

Thank you. We have rephrased the sentence according to your suggestion.

page 18, line 30: methanotrophs will also influence $CH_4$ concentrations

We think that the methanotrophic activity most likely does not cause diurnal variation in $CH_4$ fluxes, so it may not be relevant to mention here. The effects of methane oxidation have been discussed in section 4.1. where we considered potential reasons for the different $CH_4/CO_2$ ratios between dry and wet season.

General notes on formatting: there are some inconsistencies with subscripts on methane ($CH_4$) and carbon dioxide ($CO_2$) abbreviations.

Thank you. We have corrected formatting on the page 17.

---

## Author Response (AR1)

Dear Dr. Räsänen,

Your manuscript has been reviewed by three reviewers. They all find your work of some potential interest and very well written. However, the reviewers have raised quite substantial concerns that must be addressed to bring the manuscript to a publishable standard.

Particularly important are the recommendations to (1) clearly describe and fully justify all aspects of the methods used to guarantee the reproducibility of your study (e.g., number of sampling days / time points in both the wet and dry seasons); (2) discuss if differences in gas fluxes between ecosystems are attributable to the environment (grassland vs. bushland) or to the two different termite species / nest properties ; (3) discuss the limitations of the study, such as the implications of your sampling design / sample size and potential effects of seasonality / diurnal variation / spatial variability on gas fluxes, and appropriately tone down relevant conclusions.

Please, carefully consider these recommendations as well as the detailed observations provided by the three referees.

Thank you.

Yours sincerely,

Erika Buscardo

Thank you to the editor and reviewers for the constructive comments. For clarity we specify here our response regarding the three main points.

1) We have now specified the sampling dates of each mound in the supplement Table S2. Details of the diurnal measurements have been added to the method section 2.3.

2) We have improved the manuscript to consider these uncertainties. For example, in discussion we added the following text:

*Our biased sampling design, with one termite species dominating the bushland and the other the grassland site, does not allow us to reliably compare the two termite species. However, the relatively similar gas fluxes of the two species, or mound types, (Figures 4 and 6) imply that they do not exhibit major differences in terms of $CO_2$ or $CH_4$ emissions. This result is also highly expected as the two termite species are closely related (Bagine et al., 1994; Vesala et al., 2017), have very similar population sizes and compositions (Darlington, 1984), and also share their dietary niches (Vesala et al., 2022b). The highest fluxes of both gases were registered from two closed mounds of M. michaelseni in the grassland. It is important to notice that we did not obtain any information about the population sizes and, thus, these two M. michaelseni mounds may simply have housed larger termite populations with correspondingly larger fungus gardens than the studied M. subhyalinus mounds.*

3) The whole manuscript is edited regarding the study limitations. In abstract we have rephrased conclusions about the diurnal variation. In introduction now specify that our objective is to observe the diurnal and seasonal variation in mound fluxes. We have pointed out the limitations in our seasonal and diurnal sampling in the corresponding points in the discussion.

**Anonymous Referee #1**

Regarding the closed/open mound category, I am not entirely clear on what the hypotheses are for each type. What is the reasoning for measuring both, and what are the implications for gas fluxes? I could see adding a discussion point focusing on this contrast.

We thank the referee for his/her supportive comments.

To understand the influence of termite fluxes at the ecosystem scale it is necessary to measure both mound types. The two termite species (and mound types) studied here are the dominant mound-building termites and more or less equally present in the area. However, their distribution is not completely random or overlapping at a smaller scale. For example in the current study, open mound type was dominant in the bushland and the closed type in the grassland site. More detailed map of different mound types in the region is presented in the fig. 2 of the previous study (Vesala et. al., 2017). Because the two termite species are closely related and ecologically very similar we didn't expect major differences in the fluxes. However, the highly different nest ventilation could potentially have some role (e.g. related to the proportion of gases that is emitted via the mound or surrounding soils, effect of methane oxidation, etc.) which we wanted to find out. The results however did not imply that the mound type would have any significant effects on the fluxes. We have now stated this point in the discussion.

Vesala, R., Niskanen, T., Liimatainen, K., Boga, H., Pellikka, P. and Rikkinen, J.: Diversity of fungus-growing termites (Macrotermes) and their fungal symbionts (Termitomyces) in the semiarid Tsavo Ecosystem, Kenya, Biotropica, 49(3), 402–412, doi:10.1111/btp.12422, 2017.

page 3, line 17: for information on feeding guild and termite methane see Zhou, Yong ,A. Carla Staver, and Andrew B. Davies. 2023. Species-Level Termite Methane Production Rates. Ecology 104(2): e3905. https://doi.org/10.1002/ecy.3905

Thank you. We have added this reference to the sentence.

page 4, line 16: how are foraging strategies different here? Is it about what the termites are consuming, and why might different fluxes result?

We have removed this sentence as it's not appropriate point for site description.

page 15: for mean $CO_2$ and $CH_4$ fluxes of mound and soils, are soils significant? It looks like methane in soils is negligible.

The dry season soil $CH_4$ fluxes are positive at 2 m and 4-6 m distance. In contrast, the soil $CH_4$ fluxes far from termite mounds are negative throughout the year in this area (Wachiye et al., 2020). We were directly sampling the $CH_4$ concentrations every second and the soil $CH_4$ concentrations showed a clear linear trend that was higher than the gas analyzer noise.

page 15 figure 7: might be clearer to visualise this with mound and soil fluxes compared instead of just presenting soil fluxes, as this contextualizes both measurements.

Thank you for the suggestion. We have now added the mean mound fluxes to the figure.

page 16, line 8: Fungus-farming termites are thought to have higher rates of methane production: see Rouland et al. 1993, Gomathi et al 2009, Zhou et al 2023.

We edited the sentence and included the suggested information that most studied fungus-growing species have relatively high CH4 production rates.

page 18, line 26: Sentence doesn't quite make sense - "In addition to the daily cycles of CO2 and CH4 fluxes, the relationship between CO2 and CH4 showed a clear diurnal pattern"

Thank you. We have rephrased the sentence according to your suggestion.

page 18, line 30: methanotrophs will also influence CH4 concentrations

We think that the methanotrophic activity most likely does not cause diurnal variation in CH4 fluxes, so it may not be relevant to mention here. The effects of methane oxidation are discussed in sections 4.1. and 4.3. where we consider potential reasons for the different CH4/CO2 ratios between dry and wet season for the mounds and surrounding soils, respectively.

General notes on formatting: there are some inconsistencies with subscripts on methane (CH4) and carbon dioxide (CO2) abbreviations.

Thank you. We have corrected formatting on the page 17.

**Anonymous Referee #2**

Comments on the article 'Assessing CO2 and CH4 fluxes from mounds of African fungus-growing termites' from Matti Räsänen and co-authors

The article is well written, and contains interesting field data about a topic which is still relatively unknown, and of which relatively little data exists. The amount of mounds on which this story is based is small, but this is understandable considering the field conditions, and the difficulty to find suitable mounds. Also, the experiment is well designed in the sense that many environmental variables were measured. It is nice how some entomology studies are used to interpret the data.

There are a few general comments which should be addressed by the authors, followed by many detailed suggestions.

General comments

**Dependency between species, nest type and location**

The authors explain the difference between open and closed mounds in the beginning, and introduce the 2 termite species. They briefly indicate that 1 species makes closed mounds, and the other species open mounds. So, these variables (open/closed mound and species 1/species 2) are not independent. Nevertheless, this is not superclear through the paper. Looking at Table 1, termite species and mound type are listed as independent variables (separate columns), but they are not independent. Additionally to this, the authors try to compare the 2 different species in 2 different environments. But, looking at Table 1, the M. michaelseni only appears once in the bushland (out of 6 mounds), and the M. subhyalinus appears only 2 in the grassland (out of 6 mounds). While it is likely out of practical considerations that the authors decide to not look for equal sampling at each side, this should be discussed better.

So, I would advise the authors to better discuss and evaluate the fact that mound type and species are not independent and to better discuss whether observed differences between grassland and bushland are a consequence of the environment, or the fact that they were basically measuring different termite species (with different nest properties).

We thank the referee for all the helpful comments and suggestions.

In Table 1, we have merged the termite species and mound type columns to better indicate that these are not independent variables. We have added discussion point regarding the species, nest type and location:

Our biased sampling design, with one termite species dominating the bushland and the other the grassland site, does not allow us to reliably compare the two termite species. However, the relatively similar gas fluxes of the two species, or mound types, (Figures 4 and 6) imply that they do not exhibit major differences in terms of $CO_2$ or $CH_4$ emissions. This result is also highly expected as the two termite species are closely related (Bagine et al., 1994; Vesala et al., 2017), have very similar population sizes and compositions (Darlington, 1984), and also share their dietary niches (Vesala et al., 2022b). The highest fluxes of both gases were registered from two closed mounds of *M. michaelseni* in the grassland. It is important to notice that we did not obtain any

information about the population sizes and, thus, these two *M. michaelseni* mounds may simply have housed larger termite populations with correspondingly larger fungus gardens than the studied *M. subhyalinus* mounds.

**Discussion of diurnal variation**

The part about the diurnal variation is interesting, but it is difficult to judge how valid/representative the measurements are. Please add on how many days the measurements were made, and how representative 4 or 5 time points are to determine a daily variation. Also, the authors mention that the largest $CO_2$ flux coincides with the highest wind speed. But, to what extent can wind speed be a factor of influence on the $CO_2$ flux if the mound is sheltered from wind speed during the measurement? Can the authors discuss this?

We have now specified the number of nighttime measurements in the method section. The S1 mound was measured on 15 November 2016 and S2 on 7 November 2016 from evening until next day morning. During each measurement the mounds were sampled by three repeated measurements from which the standard error was calculated and presented in the Fig. 5.

The diurnal sampling time points were focused on the afternoon to morning period when there is expected transition in the convective flow of the mound and because there is a lack of measurements from these periods. The sampling was less dense during the daytime when it is likely that there is a steady increase up to early afternoon based on earlier studies. A previous study measured during 6 time points, and there is no indication of fast temporal variation in the mound flux (Jamali et al., 2011). In addition, our diurnal flux measurements trend is in agreement with the high-frequency internal CO2 concentration measurements of *M. michaelseni* mounds (Ocko et al., 2017).

We observe that the wind speed maximum coincides with the largest diurnal flux value. Despite of that, the wind speed is not expected to drive the mound fluxes which originate from the deep mound structures (termite nest and fungus gardens) not affected by the ambient conditions. Wind can, however, disrupt the flux measurements, especially in case of larger mounds that do not fit completely into the chamber. Figure S2 shows how the concentration measurement looks like when the mound is not fully covered by the chamber and wind affects the measurements.

**Discussion of differences between seasons**

The differences between dry and wet season are interesting, but please discuss it further. For example, on page 17, line 15, the decrease in $CH_4/CO_2$ ratio is mentioned, seen from dry to wet season, and possibly linked to the activity of methanotrophic bacteria. So, the authors mention that during the wet season, mounds have higher water content, so these bacteria become more active. But, you also mention (page 19, line 5) that the mounds represent a stable humidity and temperature. Is there any data or literature which actually shows that the mound humidity changes with season? Or is this 'just' an hypothesis to fit your observation? Please elaborate.

Although air humidity within the nest cavities remains constantly at a high, almost saturated, level (Luscher 1961; Agarwal 1980), we suspect that the key variable for methanotrophic bacteria, inhabiting soil structures within and around the mounds, is the soil moisture which does vary between dry and wet seasons – also in termite mounds where moisture of soil wall structures seems to be largely controlled by precipitation (Jamali et al., 2011; Chen et al. 2019). We have now added the daily mean soil moisture measurements registered from the weather station to the figure 3a that shows that surface soil moisture was clearly higher during the second than during the first campaign. The first measurements were done after long dry period and the precipitation between the campaigns was 153 mm. Although soil moisture was not measured in the studied mounds, we assume that the water content of mound walls correlate with the soil moisture values.

Agarwal, V. B.: Temperature and relative humidity inside the mound of Odontotermes obesus (Rambur) (Isoptera: Termitidae). Proc. Indian Acad. Sci. (Anim. Sci.) 89, 91–99, https://doi.org/10.1007/BF03179148, 1980.

Chen, C., Wu, J., Zhu, X., Jiang, X., Liu, W., Zeng, H., and Meng, F.-R.: Hydrological characteristics and functions of termite mounds in areas with clear dry and rainy seasons, Agriculture, Ecosystems & Environment, 277, 25–35, https://doi.org/10.1016/j.agee.2019.03.001, 2019.

Lüscher, M.: Air-Conditioned Termite Nests, Sci. Am., 205(1), 138–145, doi:10.1038/scientificamerican0761-138, 1961.

Also, it is observed by the authors that the soil $CH_4$ flux is positive in the dry season, and uptake is seen in the wet season ('The dry season mean soil CH4 flux was positive at the grassland and bushland site, whereas the mean wet season flux was nearly zero with most fluxes being negative (Fig. 7, S5 and S6')).

This is unexpected, since usually higher soil moisture (wet season) leads to $CH_4$ emission, while uptake occurs when the soil is drier. Maybe I have overlooked it, but did the authors measure soil moisture during both season, and how was it indeed drier/wetter in the wet/dry season? The graphs in the suppl material, are they for wet or dry season? There the authors seem to find a weak relation between soil moisture and $CH_4$ flux. So, why do the authors observe more soil $CH_4$ flux in the dry season? Discuss this.

The main aim of the soil CH4 flux measurements was to measure the possible influence of the termite mound to the soil flux around the mound. The soil CH4 measurements were made at 2m, 4m and 6m distances from the mound. The soil fluxes at 2m distance from the mound are higher compared to the further distances. The soil CH4 flux values at further distances are in similar magnitude to previous measurements that were made in this area far from termite mounds (Wachiye et al., 2020). This means that a small proportion of the gases leave the nest from via the adjacent soil seen here at the 2m distance.

We have added following text about this topic to the section 4.3.

*Our wet season soil CH4 fluxes around the mounds were lower than dry season fluxes. A previous study found the soil CH4 flux far from termite mounds to be near zero or negative throughout the year in this area (Wachiye et al., 2020), and thus we suspect that termites are the main source of the positive CH4 fluxes measured here. Changes in the abundance of termites in foraging networks surrounding the mounds could potentially vary seasonally. Especially during rains the foraging activity of M. michaelseni is low (Lepage, 1981), which could temporarily decrease the CH4 fluxes in surrounding soils. However, this should concurrently increase CH4 fluxes of the mounds, where the foragers need to move to. As also the mound CH4 fluxes were systematically lower during rainy season than during dry season, we interpret the observed seasonal differences in CH4 fluxes to be caused mainly by changes in soil methanotrophic activity. As already discussed above, we believe that increase in soil water content enhances microbial activity, including methane oxidation, compared to low moisture conditions in the end of the long dry season.*

We have now plotted surface soil moisture in Fig. 3a.

**Sample quantity/representativeness**

This overlaps with the previous points, but just a general comment. The authors have only studied 12 mounds, only measured during 2 moments in the year, and only once diurnal variation was studied (during x days). Again, field conditions are hard, and it is understandable that the dataset is not larger. But I would encourage the authors to consider and evaluate this in their interpretation (can a comparison be made between dry and wet season based on just 2 moments? Can a conclusion on daily variation be made based on just a few measurements?).

We have compared the results of the two measurement campaigns with potential linkages to seasonal changes in biomass of termites/fungus combs and nutritional changes driven by seasonal changes in vegetation. It is, however, true that our data originate from two relatively short field campaigns and thus the sampling does not capture all the major periods of vegetation and soil moisture changes in the area. We now point out in discussion the limitation in our seasonal sampling of the mound fluxes.

Macrotermes mounds are large compared to mounds of many other termite species, which complicated the measurements and limited the number of measurements that could be done in a limited time. In field measurements, the challenge is to find mounds that can actually be measured and the measurements are difficult to perform on large mounds and especially during night in remote locations.

As mentioned above our diurnal measurement periods were focused on the day to night and night to day transition periods which are most interesting from the gas exchange point of view. Our result shows similar trend with the high-frequency internal CO2 concentration measurements of *M. michaelseni* mounds (Ocko et al., 2017). Given this agreement in the mound CO2 flux and internal CO2 concentration our work suggests that a longer-term continuous CO2 concentration measurement with occasional mound CO2 flux measurement might be a feasible approach to estimate the continuous termite flux from the mound.
* * *
One curiosity, are there no local Kenyan researchers involved in (writing up) this research?

We agree that so-called helicopter science is problematic, and local scientists should be involved in research projects. This has indeed been a case in many of projects and publications in Africa by our group (see e.g. Räsänen et al., 2017; Wachiye et al., 2020; 2022). However, there are not always local scientists interested in specific research topics and in such cases including a local name just as token would not be ethical. Thus in this paper we happen not to have any local scientists, unlike many of our other papers.

Räsänen, M., Aurela, M., Vakkari, V., Beukes, J. P., Tuovinen, J.-P., Van Zyl, P. G., Josipovic, M., Venter, A. D., Jaars, K., Siebert, S. J., Laurila, T., Rinne, J., and Laakso, L.: Carbon balance of a grazed savanna grassland ecosystem in South Africa, Biogeosciences, 14, 1039–1054, https://doi.org/10.5194/bg-14-1039-2017, 2017.

Wachiye, S., Merbold, L., Vesala, T., Rinne, J., Räsänen, M., Leitner, S., and Pellikka, P.: Soil greenhouse gas emissions under different land-use types in savanna ecosystems of Kenya, Biogeosciences, 17, 2149–2167, https://doi.org/10.5194/bg-17-2149-2020, 2020.

Wachiye, S., Pellikka, P., Rinne, J., Heiskanen, J., Abwanda, S., and Merbold, L.: Effects of livestock and wildlife grazing intensity on soil carbon dioxide flux in the savanna grassland of Kenya, Agriculture, Ecosystems & Environment, 325, 107713, https://doi.org/10.1016/j.agee.2021.107713, 2022.
* * *
Detailed suggestions

-Page 1, Line 19: is there a white space between 'CO2' and 'and'?

The space was in subscript. This is now corrected.

-Page 4, line 12: to be result from→ to be a result of

Corrected

-Page 4, line 17: and thus may cause different fluxes-→ which may result in different fluxes

We have removed this sentence.

-Page 7: line 10-11: diurnal variation measurements were made: on how many days?

The mound S1 and S2 were measured from evening until next day morning (S1 on 15 November 2016 and S2 on 7 November 2016).

-Fig 2: add on the x-axes which method this number belongs to (Calculated Cone volume/)

Thank you. Corrected the x-axes text and specified the cone volume equation number in the caption text.

-Page 8, line 5: was→ were

Corrected

-Page 9, sentence on line 8 and 9: sentence is incorrect, check if comma or verb is missing?

Thank you. We rephrased the sentence to read "The relationship of the environmental variables to the fluxes and to the ratio of $CH_4$ to $CO_2$ flux was tested using all the measurements (n = 18)."

- Page 10: line 11-12: why are S1 and MR4 compared? They have a similar volume, but have a different species and environment, so is this comparison useful/valid?

This comparison may not be appropriate. We have removed the sentence.

- Figure 6: the legend of the triangles and circles is only given in the first figure (6a). Maybe add a sentence in the caption as well

We have now indicated this in the caption.

- Fig 6f: you plot here the standard deviation of the CO2 flux. Please add to your material and methods how you obtain the standard deviation. Is this the standard error of the linear regression slope?

We made always three repeated measurements with 5 min break in between. The standard deviation was calculated from the three repeated measurements. We have now explained this in the method section and in the Fig. 6 caption.

- Table 2: is there a white space missing between CH4 and flux?

The space was in subscript. This is now corrected.

- Page 16, line 12: mound outer dimensions correlate positively with the size….. This you take from literature, right? Although clear from the paper (you didn't count the termites), maybe clarify this to the reader (sentence below just a suggestion, feel free to ignore or improve)

*As found by previous studies, the mound outer dimensions of the Macrotermes species (of which both our species belong to) correlate positively…..*

Thank you. We rephrased the sentence and it now reads "As found by previous studies, the mound outer dimensions of both Macrotermes species correlate positively with …."

- Page 16, line 32: elaborate maybe 1 sentence what you mean with sterile, and why that leads to constant activity over the year

The paragraph was edited to clarify the idea that although the number of termites remain relatively constant, biomass changes in seasonally produced alates and fungus gardens could potentially cause some intra-annual variation in gas production.

-Page 17, line 5 and 8. $CO_2$ not written in subscript

Corrected

**Anonymous Referee #3**

Räsänen and colleagues investigate CO2 and CH4 fluxes from mounds of fungus-growing termites and adjacent soil at selected sites in Kenya. They chose plots in open grassland and bushland and investigate diurnal courses of fluxes. Results highlight the scaling of mound volume to CO2 flux density and lower CH4 emissions compared to literature values from soil and grass-feeding termites. The authors also stress the importance of active or dead mounds in the landscape for upscaling fluxes.

Overall, the paper is well written, methods are well established, the topic is of relevance for greenhouse gas emission budgets as fluxes from termites are still understudied and flux estimates are associated with high uncertainties. A few plotting issues should be addressed. I particularly enjoyed reading the discussion section.

Beside some general thoughts, here are a few points that should be considered in the revision of the manuscript:

- I noticed that the title had already been modified at an earlier stage, but does "Assessing" really fit? What's been assessed? I'd just leave the word out.

We thank the referee for all the helpful comments and suggestions.

Thank you for this suggestion. We have modified the title to read "Carbon dioxide and methane fluxes from mounds of African fungus-growing termites"

- No local collaborators as co-authors? In times where "helicopter science" is highly debated, it is hard to understand why not a single person from the region or with local expertise made it to the list of co-authors. I'm not saying that random people from the street should be chosen and I fully agree that a co-author must have made a considerable (scientific) contribution to the study, but weren't there any African institutions involved where people could have been invited to contribute? Please, at least consider this in the planning phase of any upcoming field studies abroad.

We agree that so-called helicopter science is problematic, and local scientists should be involved in research projects. This has indeed been a case in many of projects and publications in Africa by our group (see e.g. Räsänen et al., 2017; Wachiye et al., 2020; 2022). However, there are not always local scientists interested in specific research topics and in such cases including a local name just as token would not be ethical. Thus in this paper we happen not to have any local scientists, unlike many of our other papers.

Räsänen, M., Aurela, M., Vakkari, V., Beukes, J. P., Tuovinen, J.-P., Van Zyl, P. G., Josipovic, M., Venter, A. D., Jaars, K., Siebert, S. J., Laurila, T., Rinne, J., and Laakso, L.: Carbon balance of a grazed savanna grassland ecosystem in South Africa, Biogeosciences, 14, 1039–1054, https://doi.org/10.5194/bg-14-1039-2017, 2017.

Wachiye, S., Merbold, L., Vesala, T., Rinne, J., Räsänen, M., Leitner, S., and Pellikka, P.: Soil greenhouse gas emissions under different land-use types in savanna ecosystems of Kenya, Biogeosciences, 17, 2149–2167, https://doi.org/10.5194/bg-17-2149-2020, 2020.

Wachiye, S., Pellikka, P., Rinne, J., Heiskanen, J., Abwanda, S., and Merbold, L.: Effects of livestock and wildlife grazing intensity on soil carbon dioxide flux in the savanna grassland of Kenya, Agriculture, Ecosystems & Environment, 325, 107713, https://doi.org/10.1016/j.agee.2021.107713, 2022.

- Abstract, Line 20-21: The fact that there is a 35% decrease of CO2 fluxes in the wet season compared to the dry season is unexpected. Maybe already indicate here the potential reasons.

This is an unexpected result. However, we cannot not completely explain this pattern in the discussion so we have opted not to include the potential reasons in the abstract. We suggest that the contrasting seasonality between the two study sites could be related to the availability and quality of food sources that termites can utilize. Bushlands have generally more abundant and diverse food sources than grasslands where the grasses are a highly competed food source. This could potentially explain the contrasting seasonality patterns between the two habitats and the higher seasonal variance in gas fluxes observed at the grassland than at the bushland site.

- Carefully check the reference list. Not all papers cited in the text are listed.

Thank you. We have added references for Brümmer et al., 2009, Amara et al., 2020 and Jamali et al., 2011.

- Figure 1 is overall very nice. What is the source of the above-ground biomass data?

We have added the corresponding references to the caption text (Amara et al., 2020; Pellikka et al., 2018).

- Sections on gas flux measurements: What is the sensitivity of the gas flux calculation to the mound volume? Have you done some calculations and could this be considered in the uncertainty estimation? Also, how stable, i.e. "how linear" was the concentration increase? Was Equation (2) really the best fit? Could you see any saturation of the concentration increase during chamber closure? And would have probably another method for flux calculation been better?

The mound fluxes are high, and we were directly measuring the gas concentrations every second using the gas analyzer. The gas concentrations were linear and no saturation was observed in the concentrations (Figure 1 in this document). Therefore, the Eq. 2 was the best fit for the data. It is

unlikely that the flux calculation introduces large uncertainty to the flux to mound volume relationship. The results show that the dead mounds should not be considered when establishing these relationships.

[Figure]

*Figure 1. Example raw concentration measurements recorded from the MR1 mound.*

- Page 7, Line 8: How did you assure gas tightness? Was the collar smoothly inserting into the soil?

The PVC was custom made to cylindrical shape to fit the collar size. Tight rubber band was used to tighten the PVC sheet against the collar. In the field measurements, the collar was inserted to soil surface and then surrounding sand was used to insulate the collar so that there is no leak between collar and soil surface. No indication of leakage was observed in the measurements except during the measurement of the S5 mound that was excluded because mound did not fit inside the chamber (Fig. S1 and S2).

- Section 2.3: Very nice setup regarding air mixing!

Thank you.

- Section 2.4: How can nest temperatures from another year be used in this study? How comparable would they be? Isn't nest temperature correlated with air temperature, soil temperature, wind speed? Or was just the relative diurnal course taken into account?

It is true that the nest temperatures are from a previous study (Vesala et al., 2019b) and we have clearly pointed that out when presenting the results. We think presenting these mound temperature results help to set the context for the environmental conditions. The main point is presented in the Fig. 3d which can be compared to the diurnal flux measurements in Fig. 5. The soils are sandy in this area which typically have lower spatial variance in soil temperatures. Furthermore, we present data only during the period with no rainfall.

- Page 9, Line 3: "of" missing between "amount" and "woody"?

Corrected

- Figure 3 (a): Rainfall should not be plotted as time series, but as a bar graph. Here it is the daily sum.

Corrected

- Figure 3 (b) and (d): x-axes labels, ticks at 0, 6, 12, 18, 24

Corrected

- Figure 4: The caption does not really tell what the lower panel shows, although it is quite obvious. Please add. A ratio should not be shown as a bar plot, but rather as points.

Thank you. We have indicated the panels (a,b,c) and explained what each panel shows in the caption. We have plotted the ratio as a bar to help the reader compare the two sites and two seasons.

- Figure 5, x-axes labels, ticks at 6, 12, 18, 24

Corrected

---

## Referee Report (RR1)

page 1 line 26: "The stability of the mound gas fluxes over diurnal and annual scales coincides with the constant nature of the nest internal gas and thermal environment that guarantees continuously favorable conditions for the fungal symbiont."

This is saying that mound fluxes are stable, but contradicts line 22 where it is stated that there can be quite large (~30-60% increase/decrease) in seasonal flux. I'd rephrase one or the other to keep the message consistent – is the main finding that fluxes are highly variable or not?

page 2 line 2: can define methane and carbon dioxide abbreviations here

page 13 line 2: add space after period in '5).However, '

page 17 line 7: is this a small intra-annual difference in flux? I wouldn't necessarily see a 64 % decrease at the grassland and 35 % increase at the bushland as 'small' – see comment from abstract as well. Perhaps think of rephrasing this finding – it seems that season certainly plays a role, and also has differential effects depending on the species.

In the following few sentences you discuss why intra-annual variation could exist – I'd frame the finding as the fluxes do indeed vary, and this could be due to the following factors you discuss from line 9 – 24.

page 18 line 6 – A bit late to mention this, but for mounds that started off in bad shape and then were deemed dead on the second measurement, are these worth including in the study at all? They don't seem to represent fully functioning termite mounds, which is what most of the discussion throughout the paper lies around. What do we gain from including mounds that were incomparable across seasons, or partially functioning from the start?

page 19 line 3: I'd be careful interpreting the spike in flux at midnight – it's one mound and one measurement, so not a lot of data to go off of there.

page 19 line 15: "supports the interpretation" is a bit awkward phrasing, maybe it's more of a hypothesis rather than interpretation?

page 19 for section 4.3: any CO2 fluxes coming from the soil are not restricted to termite-derived: could also be from microbial activity. Therefore, I think it's more useful to consider how CH4 changes in soil going further from the mound, as that's the real indicator of termite activity. CH4 values were near zero or negative, so I'd reduce the argument that this finding supports termites having a broad soil network where gas exchange is being significantly impacted.

For further study, it could be interesting to compare methane flux across diurnal cycles for soil/mound fluxes - to see if you can support or describe when termites are out foraging based on when methane is being picked up in greater concentration from the soils. Not sure if it would be a very clean way to answer that question, but just an idea!

---

## Author Response (AR2)

page 1 line 26: "The stability of the mound gas fluxes over diurnal and annual scales coincides with the constant nature of the nest internal gas and thermal environment that guarantees continuously favorable conditions for the fungal symbiont."

This is saying that mound fluxes are stable, but contradicts line 22 where it is stated that there can be quite large (~30-60% increase/decrease) in seasonal flux. I'd rephrase one or the other to keep the message consistent – is the main finding that fluxes are highly variable or not?

Thank you for pointing out this inconsistency. Our point here was that although there seem to be some variance in fluxes taking place for example between different seasons, this variance is smaller than what has been reported from mounds of other termite species that do not cultivate fungi. For example, some wood-feeding termites in Australia showed 90% decrease in CO2 flux from wet to dry season (see reference Jamali et al. 2013 in the manuscript), which is clearly more than observed in our study. We agree than in current form in the abstract this message is not clear and thus we have removed this sentence.

page 2 line 2: can define methane and carbon dioxide abbreviations here

Corrected

page 13 line 2: add space after period in '5).However, '

Corrected

page 17 line 7: is this a small intra-annual difference in flux? I wouldn't necessarily see a 64 % decrease at the grassland and 35 % increase at the bushland as 'small' – see comment from abstract as well. Perhaps think of rephrasing this finding – it seems that season certainly plays a role, and also has differential effects depending on the species.

In the following few sentences you discuss why intra-annual variation could exist – I'd frame the finding as the fluxes do indeed vary, and this could be due to the following factors you discuss from line 9 – 24.

Thank you. We have removed the sentence stating that these changes are small. Now the discussion is focused on the factors that might cause intra-annual variation.

page 18 line 6 – A bit late to mention this, but for mounds that started off in bad shape and then were deemed dead on the second measurement, are these worth including in the study at all? They don't seem to represent fully functioning termite mounds, which is what most of the discussion throughout the paper lies around. What do we gain from including mounds that were incomparable across seasons, or partially functioning from the start?

We prefer to keep this point in the discussion. It is an important factor to take into account in the sampling of the fungus-growing termite mounds. For instance, longer term studies have to take possible mound death into account in the sampling. It's also not easy to tell from outside (without excavation or mound flux or internal concentration measurements) whether a termite colony is active or has died recently. In future studies it would be useful to have a simple method, based for

example on $CO_2$ concentration measurements, to assess the aliveness of the termite colony before measuring fluxes. Not knowing the active/dead mound status makes it difficult to understand the factors affecting mound fluxes and may lead to wrong conclusions.

page 19 line 3: I'd be careful interpreting the spike in flux at midnight – it's one mound and one measurement, so not a lot of data to go off of there.

Thank you. We have removed the discussion regarding the spike in the midnight due to the limited data.

page 19 line 15: "supports the interpretation" is a bit awkward phrasing, maybe it's more of a hypothesis rather than interpretation?

Changed to "hypothesis".

page 19 for section 4.3: any CO2 fluxes coming from the soil are not restricted to termite derived: could also be from microbial activity. Therefore, I think it's more useful to consider how CH4 changes in soil going further from the mound, as that's the real indicator of termite activity. CH4 values were near zero or negative, so I'd reduce the argument that this finding supports termites having a broad soil network where gas exchange is being significantly impacted.

Both, the soil $CH_4$ and $CO_2$ fluxes were always higher close to the mound (at 2 m distance from the mound perimeter) compared to further distances (4 to 6 m from the mound perimeter) (Fig. 7). This result was clear and differed from the earlier measurements in soil feeding termites in Amazon which had enhanced soil fluxes around the mound only up to 0.5 m from the perimeter of the mound (van Asperen et al., 2021). We have discussed how the size and structure of mounds built by fungus-growing termites compared to those of soil feeding termites could potentially explain these differences. It is also well-documented that *Macrotermes* mounds in Kenyan grasslands are typically surrounded by extensive tunnel networks (Darlington 1982). We believe that such tunnels increase soil porosity and could partially explain the enhanced soil gas fluxes in mound immediate vicinity. Instead, the last paragraph related to potential differences in foraging tunnels in grasslands and bushlands is largely speculative as we have no data that could support these interpretations. We decided to delete this last paragraph.

For further study, it could be interesting to compare methane flux across diurnal cycles for soil/mound fluxes - to see if you can support or describe when termites are out foraging based on when methane is being picked up in greater concentration from the soils. Not sure if it would be a very clean way to answer that question, but just an idea!

Thank you. This is a good suggestion for a future study.

**Anonymous Referee #2**

The authors have improved the parts indicated, and the manuscript looks good. I have only three mini suggestions to improve the manuscript:

P3, line 27: during dry and rainy seasons-→ during a period in the dry and the wet season.

Corrected

P 10, line 6-9: add units between brackets for all variables, so also for P, A, dC/dt

Corrected

P 25, line 19, CH4 is not in subscript

Corrected

---

## Author Response (AR3)

Dear Dr. Matti Räsänen,

Thank you for sending the revised version of your manuscript. I am happy with the way you addressed the reviewers' comments. There remain just few minor points that should be addressed.

Page 3, L15: substitute 'consequences to the gas fluxes' with 'consequences for the gas fluxes'

Corrected

Page 8, L3: substitute 'was attach' with 'was attached'

Corrected

Page 10, L11: check the inconsistency between value cited in the Abstract / Discussion and the Results for bushlands.

Thank you. The value in the results was the correct one.

Page 13, L13: revise sentence. Points may be described as scattered; the relationship is either statistically significant or not. If significant, it can be weaker or stronger in a comparison.

Corrected

Page 13, L21: see previous comment

Corrected

Page 18, L15: substitute 'was limited one' with 'was limited to one'

Corrected

Figure legend / Table captions. They have to be self-standing, i.e. the reader needs to understand their content without reading the main text.

Thank you. We have edited the legends and captions to be self-standing.

Table 1: define termite genus in the caption

Corrected

Thank you.

Sincerely,

Erika Buscardo